# Ischemia and reperfusion injury in superficial inferior epigastric artery-based vascularized lymph node flaps

David P. Perrault[1☯], Gene K. Lee[1☯], Antoun Bouz[1☯], Cynthia Sung[1], Roy Yu[1], Austin J. Pourmoussa[1], Sun Young Park[1], Gene H. Kim[2], Wan Jiao[1], Ketan M. Patel[1], Young-Kwon Hong[1], Alex K. Wong[1]*

**1** Division of Plastic and Reconstructive Surgery and Department of Surgery, Keck School of Medicine of USC, Los Angeles, California, United States of America, **2** Departments of Pathology and Dermatology, Keck School of Medicine of USC, Los Angeles, California, United States of America

☯ These authors contributed equally to this work.
* Alex.Wong@med.usc.edu

**Data Availability Statement:** All relevant data are within the manuscript.

**Funding:** Funded by Alex Wong - 1K08HL132110-01A1. Young Kwon Hong - 5R01HL141857 &

## Abstract

Vascularized lymph node transfer (VLNT) is a promising treatment modality for lymphedema; however, how lymphatic tissue responds to ischemia has not been well defined. This study investigates the cellular changes that occur in lymph nodes in response to ischemia and reperfusion. Lymph node containing superficial epigastric artery-based groin flaps were isolated in Prox-1 EGFP rats which permits real time identification of lymphatic tissue by green fluorescence during flap dissection. Flaps were subjected to ischemia for either 1, 2, 4, or 8 hours, by temporarily occluding the vascular pedicle. Flaps were harvested after 0 hours, 24 hours, or 5 days of reperfusion. Using EGFP signal guidance, lymph nodes were isolated from the flaps and tissue morphology, cell apoptosis, and inflammatory cytokines were quantified and analyzed via histology, immunostaining, and rtPCR. There was a significant increase in collagen deposition and tissue fibrosis in lymph nodes after 4 and 8 hours of ischemia compared to 1 and 2 hours, as assessed by picrosirius red staining. Cell apoptosis significantly increased after 4 hours of ischemia in all harvest times. In tissue subject to 4 hours of ischemia, longer reperfusion periods were associated with increased rates of CD3[+] and CD45[+] cell apoptosis. rtPCR analysis demonstrated significantly increased expression of CXCL1/GRO-α with 2 hours of ischemia and increased PECAM-1 and TNF-α expression with 1 hour of ischemia. Significant cell death and changes in tissue morphology do not occur until after 4 hours of ischemia; however, analysis of inflammatory biomarkers suggests that ischemia reperfusion injury can occur with as little as 2 hours of ischemia.

## Introduction

Lymphedema is a chronic progressive disease characterized by tissue swelling and lymph fluid stasis that ultimately leads to tissue fibrosis, limb disfigurement, and loss of function. In the United States, lymphedema most commonly occurs following oncological therapy, such as

5R01DK114645. The funders had no role in study design, data collection and analysis, decision to publish, or preparation of the manuscript.

**Competing interests:** The authors have declared that no competing interests exist.

surgery or radiation [1–4]. There is no curative treatment for lymphedema, but vascularized lymph node transfer (VLNT) has been shown to be an effective surgical treatment modality for secondary lymphedema [5–13]. The success of VLNT is dependent on the survival of lymph nodes within transplanted tissue, as they are the functional units responsible for fluid transport.

The signs of ischemia in vascularized lymph nodes are less readily appreciable than, for example, muscle flaps. The lymph nodes are often encased in a cushion of fat, which has the potential to mask subtle visual signs of lymph node ischemia or ischemia injury. Furthermore, other than partial or complete flap loss, the success or failure of a VLNT is a long-term clinical outcome and not known in the immediate post-operative period. Therefore, the perioperative appearance of VLNT flaps may not be sensitive enough to correlate with outcome. As such, knowing how lymphatic tissue responds to ischemia-reperfusion injury and defining a critical ischemia time would be clinically valuable.

Ischemia is defined as a decrease or absence of blood delivery to tissue [14]. Critical ischemia time is the maximum amount of time a tissue can withstand ischemia without evidence of injury [15]. Most tissues can tolerate ischemia without significant injury if the ischemic period is relatively short. During free tissue transfer, ischemia time can range from 45 minutes to 3 hours [16]. Ischemia time is a potentially modifiable factor influencing the long-term viability of transplanted tissue. While a short period of ischemia is inevitable during free tissue transfer, prolonged ischemia initiates a cascade of cellular changes that result in irreversible cell membrane damage, which translates to poor post-operative outcomes. In addition to the deleterious effects of prolonged ischemia, organ reperfusion also accelerates the rate of cellular damage that leads to apoptosis [17].

Critical ischemia time in muscle, skin, and other tissues have been extensively investigated in animal and human models [15, 18–20]. Relatively little is known about the effect of ischemia and ischemia-reperfusion injury on lymph nodes and lymphatic tissue [21, 22]. Since we recently developed and characterized a transgenic rat which facilitates *in situ* real time visualization of lymphatic tissue (Prox1-EGFP), this has enabled us to precisely identify and isolate lymph nodes from surrounding tissue and analyze the cellular changes that occurred within lymph nodes [23]. Here, we conducted a detailed study on the effect of ischemia and ischemia-reperfusion on lymph nodes, investigating the biochemical and physiological changes of lymph nodes in response to ischemia-reperfusion injury.

## Materials and methods

### Animals

All animal experiments were approved by the Institutional Animal Care and Use Committee of the University of Southern California. The Prox1-EGFP reporter rats used in this study were developed previously, characterized our laboratory, and described by Jung, E. et. al. [23], Specifically, a Prox1-harboring BAC (RP23-360I16), where the EGFP gene was inserted distal to the mouse Prox1 proximal promoter was used to generate a Sprague-Dawley Prox1-EGFP founder line at Cyagen Biosciences and subsequently expanded at our facility. In this experiment, forty-eight adult Prox1-EGFP rats weighing between 300 and 500 grams, were used. The animals were housed in a temperature and light controlled environment and fed a standard rodent diet and water ad libitum. Hydrogel and nutrient rich gel were provided postoperatively. The animals were observed twice daily, their behavior was assessed for signs of distress, and their wounds checked for infection, dehiscence, and other complications. At the experimental endpoint, animals were euthanized by carbon dioxide asphyxiation followed by a confirmatory double thoracotomy.

## Operative procedure

Rats were anesthetized with 1mL/100g of a ketamine/xylazine cocktail (100mg/kg ketamine + 10mg/kg xylazine) via intraperitoneal injection and were given approximately 1mg/kg slow release buprenorphine subcutaneously immediately prior to surgery. Epigastric flaps were created as previously described [23, 24].

Briefly, the abdomen was shaved, a depilatory cream was applied, the area was prepped with povidone-iodine scrub solution, and a 2–3 cm skin incision was made in line with the inguinal crease. The subcutaneous tissue flap was dissected free from the skin, verified to contain lymph nodes by green fluorescent signal, and further isolated based on the superficial inferior epigastric artery and vein (SIEA and SIEV). The SIEA and SIEV were then skeletonized under a surgical microscope (M500N; Leica Microsystems, Wetzlar, Germany) down to their origin off the femoral vessels. All perforating vessels were either ligated or cauterized to ensure that the vascular pedicle consisted of only the SIEA and SIEV. The femoral artery and vein distal and proximal to the SIEA and SIEV were then skeletonized (Fig 1A), and microsurgical vessel clamps were placed on the femoral vessels proximal and distal the SIEA and SIEV (Fig 1B) to achieve total ischemia. The flap was placed back into the animal, and the incision was closed with simple interrupted 4–0 Vicryl (Ethicon) sutures. This procedure was then repeated on the contralateral side.

## Tissue processing and staining

Lymph node specimens were harvested and fixed in 10% neutral buffered formalin for 24 hours, embedded in paraffin, and stained with hematoxylin & eosin and Picrosirius red. A double-staining assay was also performed using Terminal deoxynucleotidyl transferase (TdT) dUTP Nick-End Labeling (TUNEL) assay and immunofluorescence staining. Briefly, lymph node sections were deparaffinized, underwent antigen retrieval using a sodium citrate buffer, permeabilized, and then incubated with anti-CD3 and anti-CD45 overnight (1:200 rabbit anti-rat; ab5690, ab10558, Abcam). Tissues were then washed in PBS and incubated with red secondary antibodies for CD3 and CD45 (1:200 goat anti-rabbit Alexa Fluor 594, Thermo Fisher Scientific, Massachusetts, USA). Next, the In-Situ Cell Death Detection Kit, Fluorescein (Sigma-Aldrich, USA) was used according to the manufacturer protocol. Finally, sections were mounted and stained with DAPI.

Collagen content within the harvested lymph nodes was quantified with Picrosirius red staining. Samples were deparaffinized, rehydrated with distilled water, then incubated in Picrosirius red solution (Sigma-Aldrich, USA) for one hour. The samples were then washed in an acid bath, dehydrated using absolute alcohol, and cleared in a xylene bath. Images were obtained and quantified to assess the degree of tissue fibrosis.

## Image analysis

The images of the double-staining assay were obtained with a Leica TCS SP8 confocal microscope, and the immunofluorescence stained slides were analyzed using ImageJ (National Institutes of Health, Bethseda, MD). For each analysis, three high-power fields per section were randomly selected and then analyzed by a blinded reviewer. First, the number of TUNEL-positive cells per square millimeter was quantified. Next, the percentage of apoptotic CD3[+] cells was calculated by dividing the number cells positive for both CD3 and TUNEL by the total number of CD3[+]. The same analysis was performed for CD45[+] cells.

The images of the Picrosirius red staining were obtained with a Keyence BZ-X700 microscope (Itasca, IL), and the samples were analyzed using ImageJ (National Institutes of Health, Bethseda, MD). A custom macro was written in ImageJ Macro language to calculate the

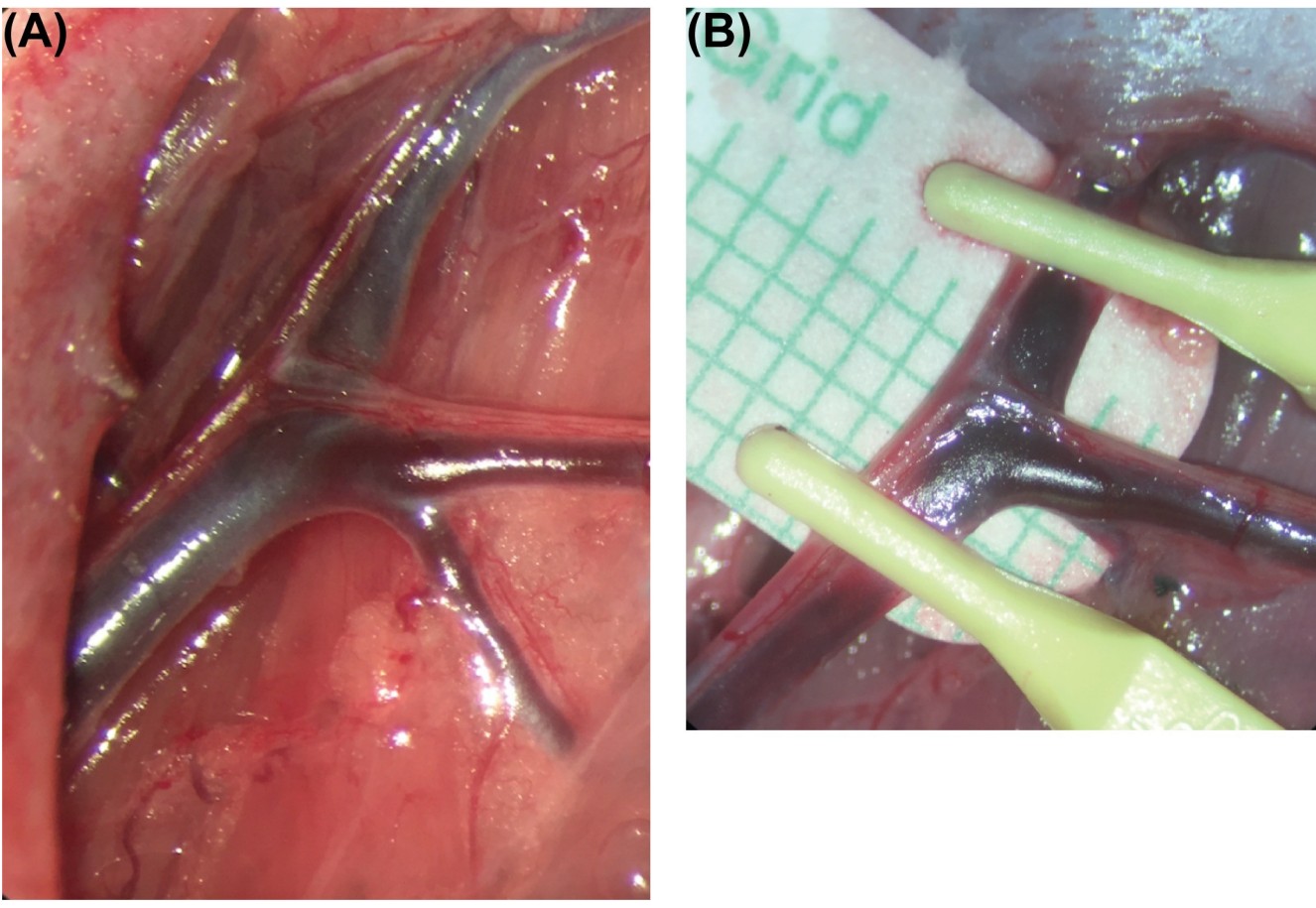

**Fig 1. Blood supply and ischemia induction of the groin flaps.** (A) Gross image of the skeletonized vessels supplying the groin flap created. (B) Gross image depicting clamping of the distal and proximal femoral vascular bundle to achieve complete ischemia in the groin flap.

percent area of collagen for tissue stained with Picro-Sirius Red by using the Color Threshold function in ImageJ (Appendix A: Picrisirius_red_ijm). The macro determines the red pixels in an image by using threshold settings of hue, saturation, and brightness. The threshold settings were determined by using a test set of 106 images. The macro then converts the red pixels into individual particles and calculates the percent area that is stained. The percentage of collagen area is calculated as the area of Picrosirius red stained tissue divided by the total area in order to quantify tissue fibrosis.

### rtPCR

Tissue collected for rtPCR was placed into cryovials and flash frozen in liquid nitrogen. The frozen tissue was ground with a mortar and pestle, and total RNA was extracted using the TRIzol reagent (Invitrogen, Carlsbad, CA, USA) according to the manufacturer's protocol. 1 μg of total RNA was used to construct cDNA using the QuantiTect Reverse Transcription kit (Qiagen, Frederick, Maryland, USA). Real time quantitative PCR (qPCR) was performed with the PerfeCTa SYBR Green Supermix (Quanta Biosciences, Inc., Gaithersburg, Maryland, USA) using the CFX96 Real-time PCR Detection System (Bio-Rad Laboratories, Hercules, CA). The specificity of the amplification reactions was monitored by melting curve analysis. The threshold cycle (Ct) value for each gene was normalized to the Ct value for GAPDH.

The specific primers used were: 5'-GGTGTGAACGGATTTGGCCG-3' forward and 5'-GTGCCGTTGAACTTGCCGTG-3' reverse for Glyceraldehyde 3-phosphate dehydrogenase (GAPDH), 5'-GCTGCTCACCATGCTGCTCT-3' forward and 5'-CAAGGCACTGCAGGGT CAGT-3' reverse for Platelet endothelial cell adhesion molecule (PECAM-1), 5'-TTGCC TTGACCCTGAAGCCC-3' forward and 5'-AGCGTTCACCAGACAGACGC-3' reverse for CXCL1/GRO-α, 5'-TCTCATTCCTGCTCGTGGCG-3' forward and 5'-GGGCTACGGGC TTGTCACTC-3' reverse for Tumor necrosis factor-α (TNF-α), and 5'-xCCTACCACACT CACGGACGC-3' forward and 5'-CCACAGCCGGGTTGGTGTAA-3' reverse for Mucin-1 (MUC1).

## Statistics

Normality of the study sample was confirmed using the Anderson-Darling test. One-way ANOVA with Tukey's multiple comparisons tests were used to determine mean differences between groups, and statistical significance was established using a cut-off of $p < 0.05$. All analyses and graphs were generated in GraphPad PRISM8 (GraphPad Software, Inc., La Jolla, CA), and values were reported as mean ± standard deviation.

## Results

### Ischemia and reperfusion of vascularized lymph node flaps

Lymph node flaps based on the superficial inferior epigastric artery and vein (SIEA/V) as described in the methods section were elevated and rendered ischemic by atraumatic occlusion of the femoral artery and vein proximal and distal to the pedicle for 1, 2, 4, or 8 hours (n = 12 rats per ischemic period; n = 48 total), after which the incision was reopened, and the microvascular clamps were removed to allow vascular reperfusion (Fig 2). The flaps were placed back into the animal, wounds were closed, and a post-operative bandage with anchor tape was applied to prevent the animal from disturbing the flap, while preserving the ability to ambulate.

At 0 hours (control), 24 hours, and 5 days lymph nodes within the epigastric flaps were precisely visualized, using GFP fluorescence microscopy (MZ10F; Leica Microsystems, Wetzlar, Germany), dissected free from the surrounding adipose tissue and harvested for further analysis (n = 4 per time point).

### Gross tissue analysis

To the unaided eye, the flaps demonstrated changes consistent with ischemia and reperfusion injury (Fig 3). Flaps harvested 24 hours or 5 days after reperfusion were more edematous than those harvested immediately. Tissue subject to 8 hours of ischemia followed by reperfusion were firm to the touch and ecchymotic, which was not seen in other groups.

### Hematoxylin and eosin staining analysis

In order to asses ischemia-induced morphological changes, whole lymph nodes were cross-sectioned and stained with hematoxylin and eosin. Analysis of the stained tissues revealed preservation of the nodal structure with 1 and 2 hours of ischemia, but significant fibrotic change occurred after 4 and 8 hours of ischemia (Fig 4). Lymph nodes subject to 4 and 8 hours of ischemia exhibited increased areas of fat deposition, fibrosis, and decreased cellularity.

### Picrosirius red staining analysis

To quantify fibrosis in response to ischemia-reperfusion injury, lymph nodes were stained with Picrosirius red. Collagen content increases with ischemia time (Fig 5A). Specifically, the 4

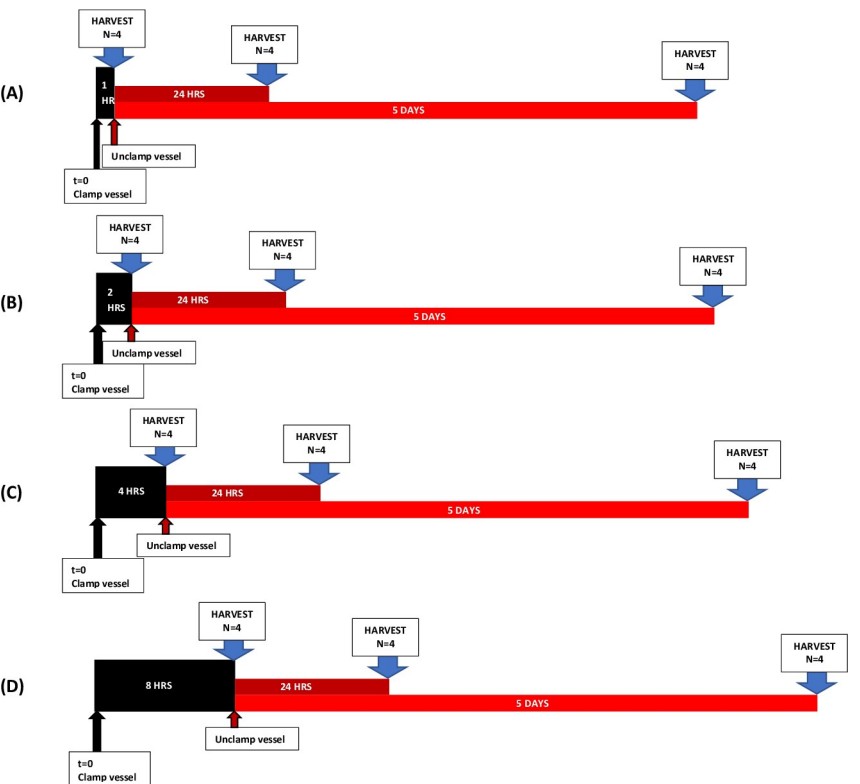

**Fig 2. Schematic of experimental groups.** Timeline depicting induction of ischemia (black arrows), re-establishment of blood flow (red arrows), and time of harvest (blue arrows) in groups subject to (A) 1 hour (B) 2 hours, (C) 4 hours, and (D) 8 hours of ischemia.

hour to 8 hour ischemia time groups showed significantly increased percentage of collagen compared to the 2 hour group (Fig 5B).

## Immunofluorescence staining analysis

We measured the rate of cell apoptosis using the TUNEL assay (Fig 6). Between 2 and 4 hours of ischemia, the rate of cellular apoptosis increased from 1264 cells/mm$^2$ to 6024 cells/mm$^2$ (Fig 7A, P<0.0001). The overall apoptosis rate increased within 24 hours of reperfusion and continued to rise with time. In flaps subject to 1 hour of ischemia, 24 hours of reperfusion was associated with an increase in apoptotic cell density. Specifically, apoptotic cell density was 977 cells/mm$^2$ at 0 hours of reperfusion and 2460 cells/mm$^2$ at 24 hours of reperfusion (Fig 8A, P<0.01). A reperfusion time of 5 days was associated with a dramatic increase in the apoptotic cell density, 5324 cells/mm$^2$ (P<0.0001). Flaps subject to 2 hours of ischemia displayed an almost identical trend. Twenty-four hours of reperfusion was associated with a total apoptotic cell density of 3064 cells/mm$^2$, compared to 1265 cells/mm$^2$ at 0 hours of reperfusion (Fig 8B, P<0.01). Similarly, 5 days of reperfusion was associated with in elevated apoptotic cell density (5441 cells/mm$^2$, P<0.0001). Flaps subject to 4 and 8 hours of ischemia displayed fairly high rates of apoptosis that did not increase significantly with longer reperfusion times.

Examination of TUNEL stained lymph nodes demonstrated that the majority of apoptotic cells were localized to the periphery and cortical areas of the lymph nodes (Fig 9). Based on the anatomy and composition of lymph nodes, we hypothesized that these apoptotic cells were likely B and T cells [25]. To test this hypothesis, we used antibodies for CD3, which is specific for T-cells, and CD45, which is a pan-leukocyte marker (Figs 10 and 11). In tissues harvested

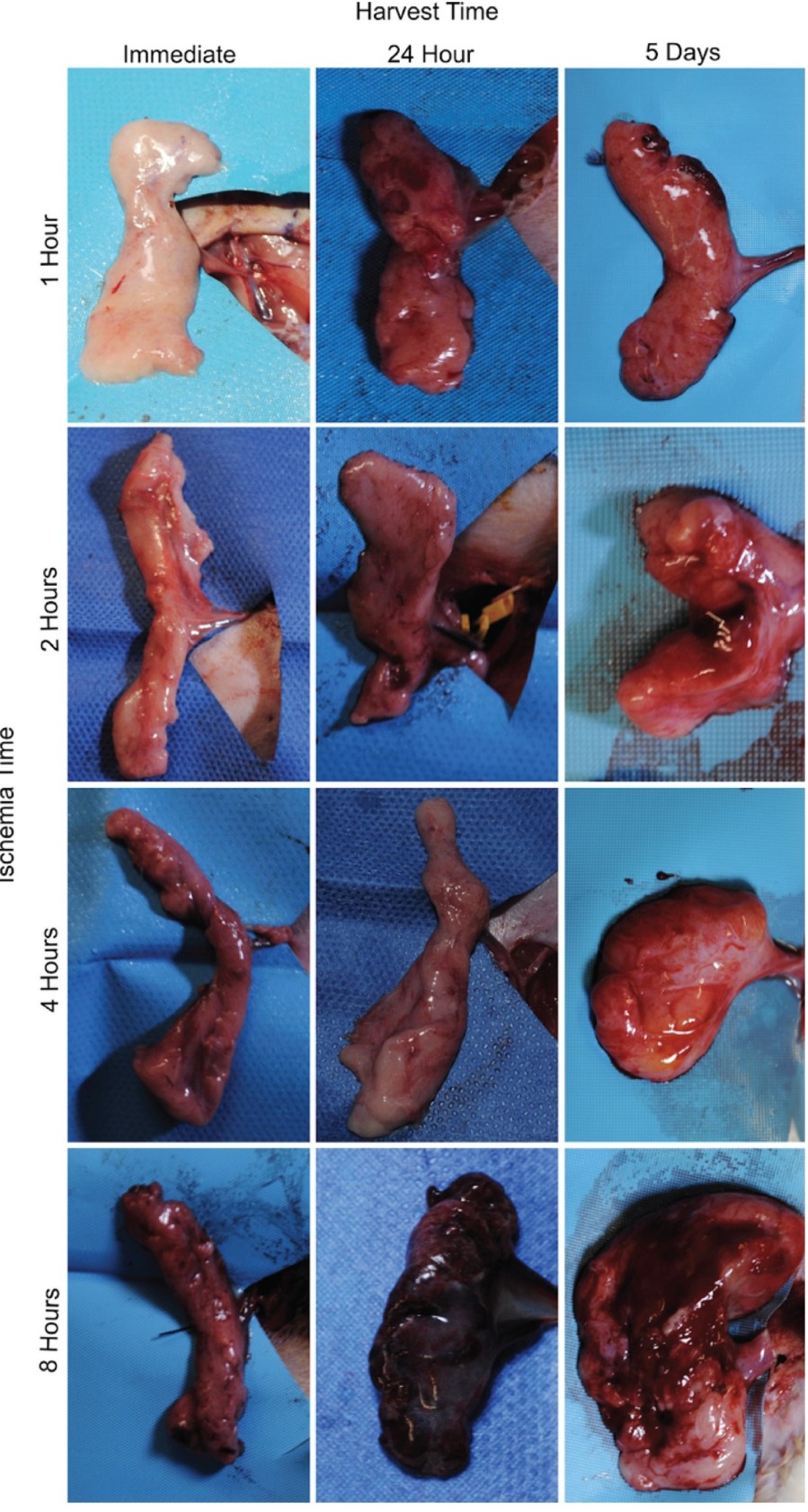

**Fig 3. SIEA flaps grossly demonstrated greater ischemic change with increasing ischemia time.** Gross photos depicting morphological changes that occur in groin flaps after exposure to various periods of ischemia and reperfusion. As ischemia and reperfusion times increase, the tissue exhibits signs of ischemic injury.

immediately, cellular apoptosis became significant after 2 hours of ischemia had passed. However, the rate of apoptosis in leukocytes and T cells demonstrated a more delayed effect. Apoptotic leukocytes and T cell levels remained relatively low at 1, 2, and 4 hours of ischemia but dramatically increased at a time point between 4 and 8 hours of ischemia. In the flaps harvested immediately, when the ischemic period was increased from 4 hours to 8 hours, the percentage of apoptotic T cells increased from 9.3% to 72.6% (Fig 7B, P<0.0001) while the percentage of apoptotic leukocytes increased from 12.7% to 84.5% (Fig 7C, P<0.0001).

In tissues subject to 4 hours of ischemia, apoptosis also increased considerably within 24 hours of reperfusion. The percentage of apoptotic T cells increased from 9.31% to 19.22% (Fig 12B, P<0.001). Similarly, the percentage of apoptotic leukocytes increased from 12.76% to 33.85% but then dramatically increased to 87.62% after 5 days of reperfusion (Fig 12C, P<0.0001). Interestingly, the rates of overall apoptosis did not show any significant changes with longer periods of reperfusion; all apoptosis rates were fairly high for tissues harvested immediately, after 24 hours, and after 5 days.

## qRT-PCR

Significant molecular changes occur in response to ischemia [20]. Based on a literature review, we chose to test a panel of biomarkers that have been previous implicated in ischemia and

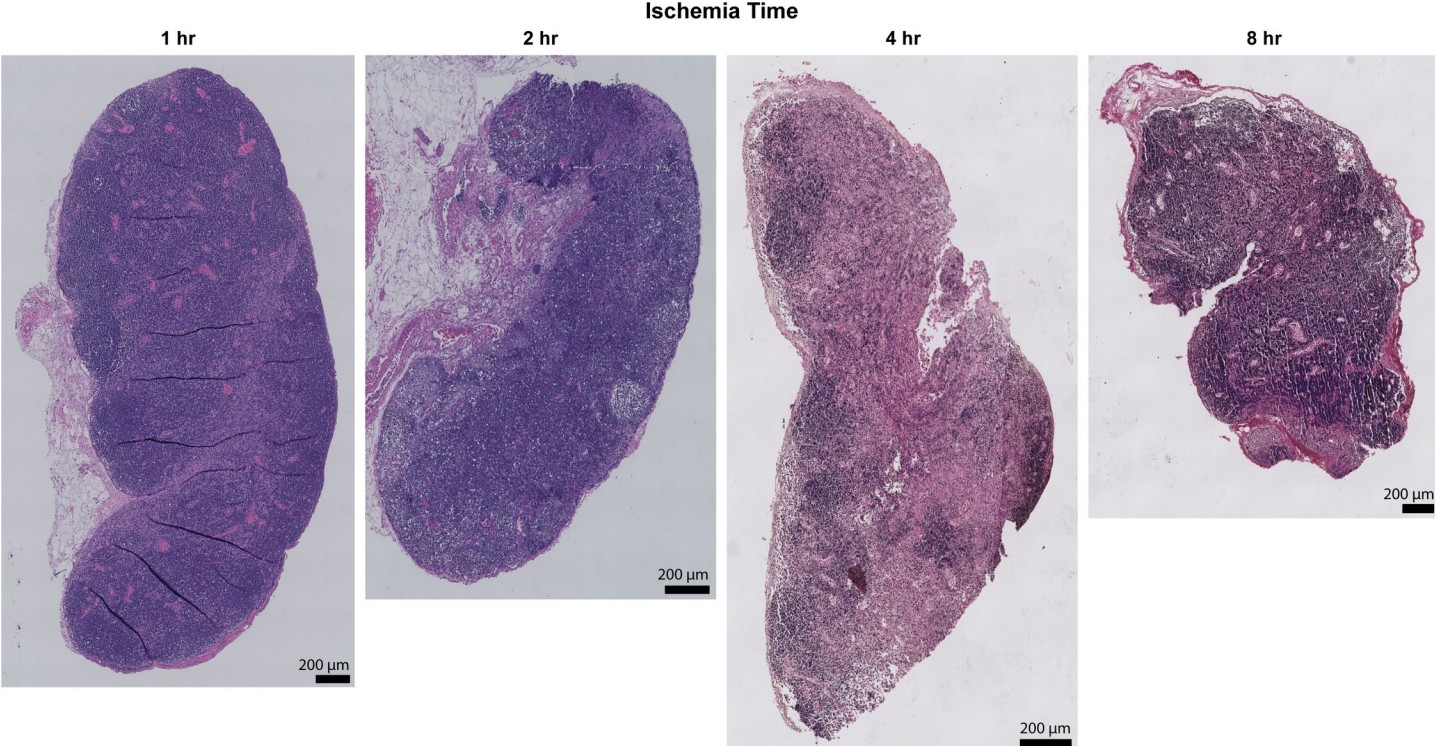

**Fig 4. Hematoxylin and eosin staining analysis of lymph node cross-sections.** Representative images of lymph node cross-sections stained with hematoxylin and eosin after various ischemia times. There is progressive nodal infarction with loss of viable lymphocytes. Coagulation necrosis is present at 4 hours and structural changes to the lymph node at 8 hours with loss of architecture and contraction. X20 magnification, x1.0 zoom. Scale bar = 200 µm.

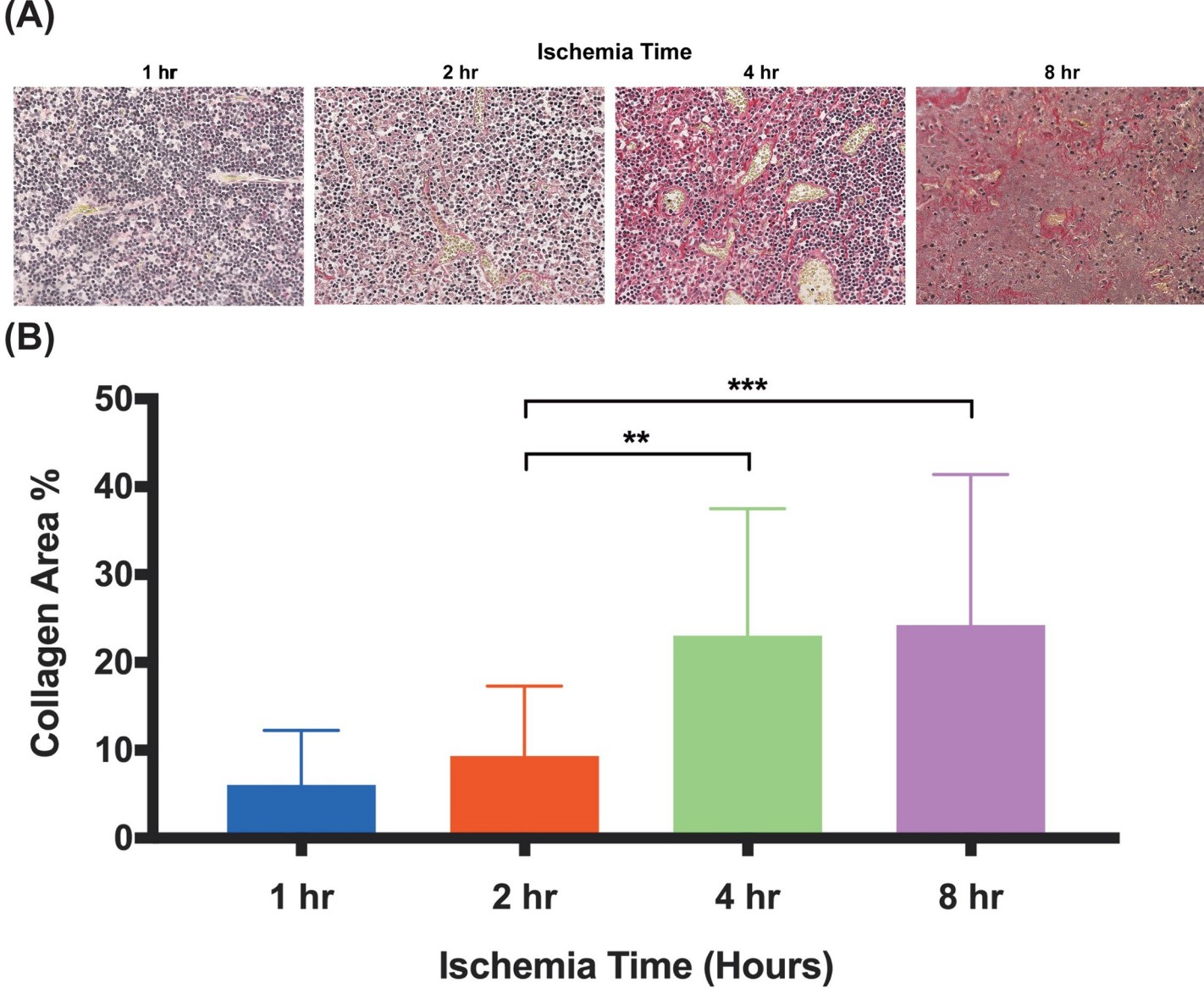

**Fig 5. Lymph node collagen content increases significantly at 4 and 8 hours of ischemia.** (A) Representative microphotographs of Picrosirius red stained lymph nodes after varying periods of ischemia, all harvested immediately after reperfusion. (B) Quantitative analysis of collagen content in lymph nodes in three random high-powered fields per lymph node. Data are presented as mean ± S.E.M. N = 12 lymph nodes per group. *p<0.05, **p<0.01, ***p<0.001.

ischemia reperfusion injury [26–33]. qRT-PCR analysis demonstrated significant changes in gene expression for all biomarkers measured. Although elevated gene expression was observed after 1 hour of ischemia in tissue harvested immediately after reperfusion, statistical significance was not reached until 2 hours of ischemia. Flaps subject to 2 hours of ischemia demonstrated upregulated CXCL1/GRO-α (40x; Fig 13A), PECAM-1 (3x; Fig 14A), and MUC1 (5x; Fig 15A). After 4 hours, upregulation of CXCL1/GRO-α (22x; Fig 13A), PECAM-1 (3x; Fig 14A), MUC1 (6x; Fig 15A), and TNF-α (3x; Fig 16A) was observed.

In tissue harvested 24 hours after reperfusion, significant changes in gene expression were seen as early as after 1 hour of ischemia: PECAM-1 (3x; Fig 14B), MUC1 (6x; Fig 15B), and TNF-α (3x; Fig 16B). Tissue subject to 2 hours of ischemia demonstrated upregulation of

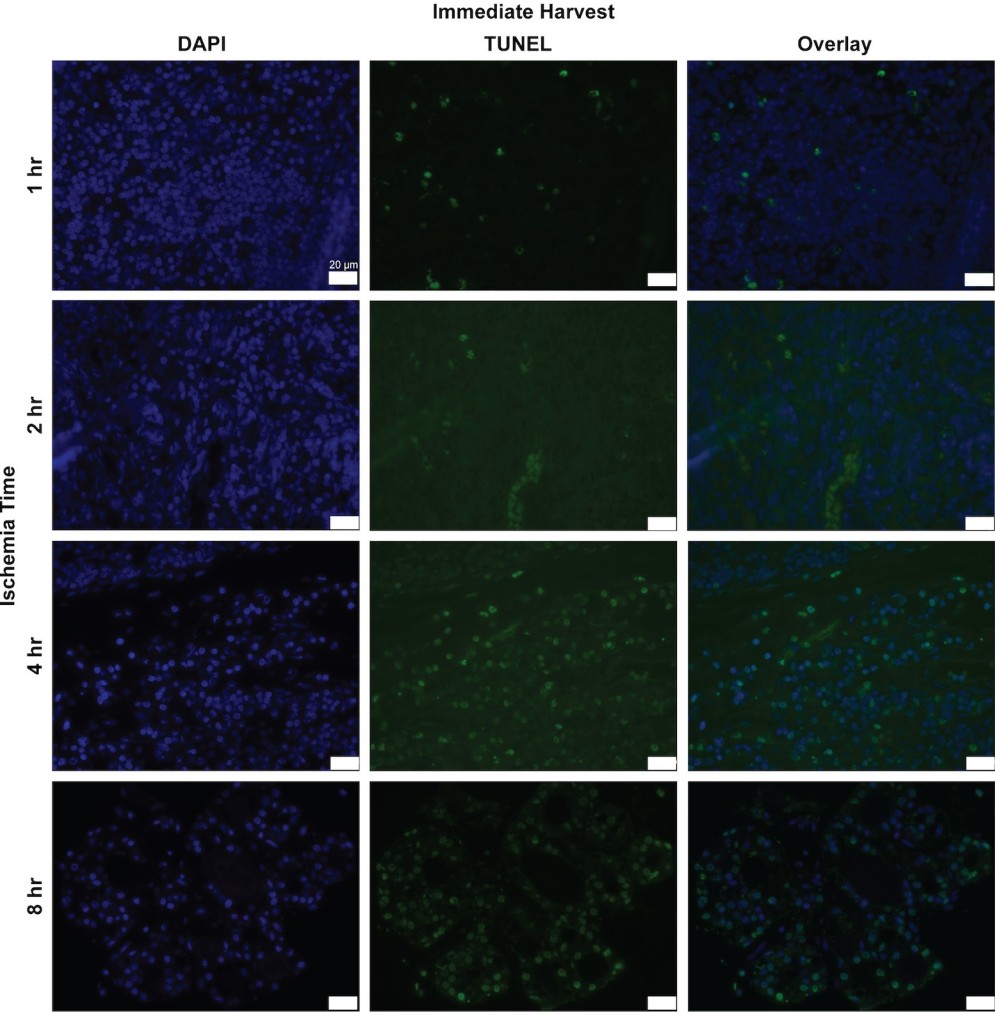

**Fig 6. Histologic and immunofluorescence analysis for TUNEL in rat lymph node cross-sections.** Representative images of lymph node cross sections of TUNEL staining (green) with DAPI (blue) after various ischemia times with immediate harvest. x63 magnification, x1.0 zoom. Scale bar = 20 um.

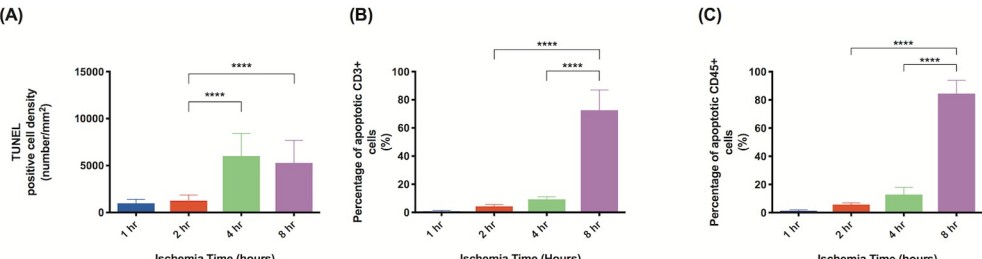

**Fig 7. Longer ischemia time increases frequency of cell apoptosis.** Quantitative analysis of cells staining positive for: (A) TUNEL (all apoptotic cells); (B) TUNEL and CD3 (apoptotic T cells); (C) TUNEL and CD45 (apoptotic leukocytes). All samples were from tissue harvested immediately after reperfusion was established. Analysis was completed in three random high-powered fields per lymph node. Data are presented as mean ± S.E.M. N = 12 lymph nodes per group. $^{*}p<0.05$, $^{**}p<0.01$, $^{***}p<0.001$, $^{****}p<0.0001$.

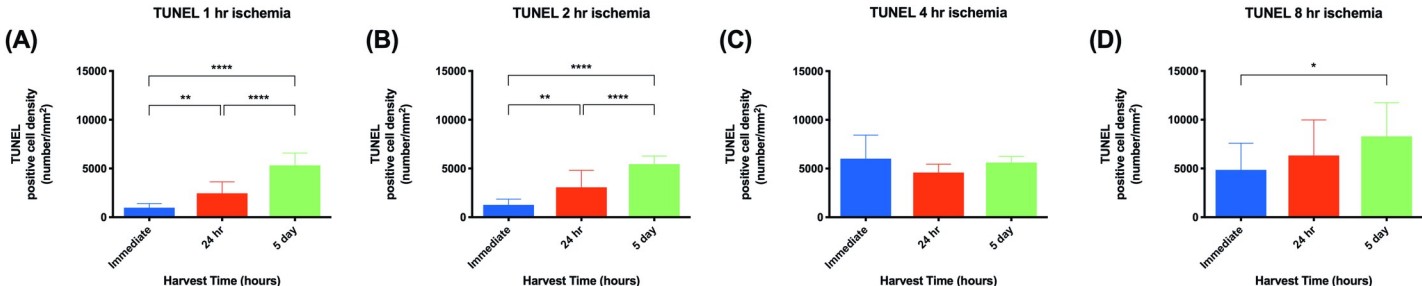

**Fig 8. Apoptotic cell density increases with increasing harvest time.** Quantitative analysis of the density of apoptotic cells in lymph nodes subject to (A) 1-hour, (B) 2-hours, (C) 4-hours, and (D) 8-hours of ischemia. There is a trend of increasing apoptotic cell density as harvest time increases. Analysis was completed in three random high-powered fields per lymph node. Data are presented as mean ± S.E.M. N = 16 lymph nodes per group. *p<0.05, **p<0.01, ***p<0.001, ****p<0.0001.

CXCL1/GRO-α (27x; Fig 13B), PECAM-1 (4x; Fig 14B), MUC1 (10x; Fig 15B), and TNF-α (4x; Fig 16B). There were no significant differences in gene expression following 8 hours of ischemia.

CXCL1/GRO-α levels returned to baseline in ischemic tissue that had been allowed 5 days of reperfusion, whereas levels of PECAM-1, MUC1, and TNF-α in the same tissue remained elevated (Figs 13C, 14C, 15C and 16C). After 1 hour of ischemia, upregulation of PECAM-1 (2x, Fig 14C) was observed. After 2 hours of ischemia gene expression was significantly elevated in: PECAM-1 (4x, Fig 14C), MUC1 (5x, Fig 15C), and TNF-α (6x, Fig 16C).

## Discussion

Prolonged ischemia and subsequent ischemia-reperfusion injury as the potential to significantly impact outcomes in free tissue transfer. Although the critical ischemia time of most tissue types has been comprehensively described, we have a limited understanding of the effect ischemia has on lymph nodes [15, 18, 19, 34]. Vascularized lymph node transfer has emerged as a viable option for the treatment of early lymphedema. The physical appearance of fat and lymphatic tissue is likely not a reliable marker for small but clinically significant ischemia-reperfusion injury. Additionally, a successful outcome from a VLNT is often determined long after the initial surgery. Given these unique characteristics of VLNT, understanding how lymph nodes tolerate ischemia and even defining a critical ischemia time would be highly useful information for clinical practice. We aimed to quantify the apoptosis and fibrosis, as well as

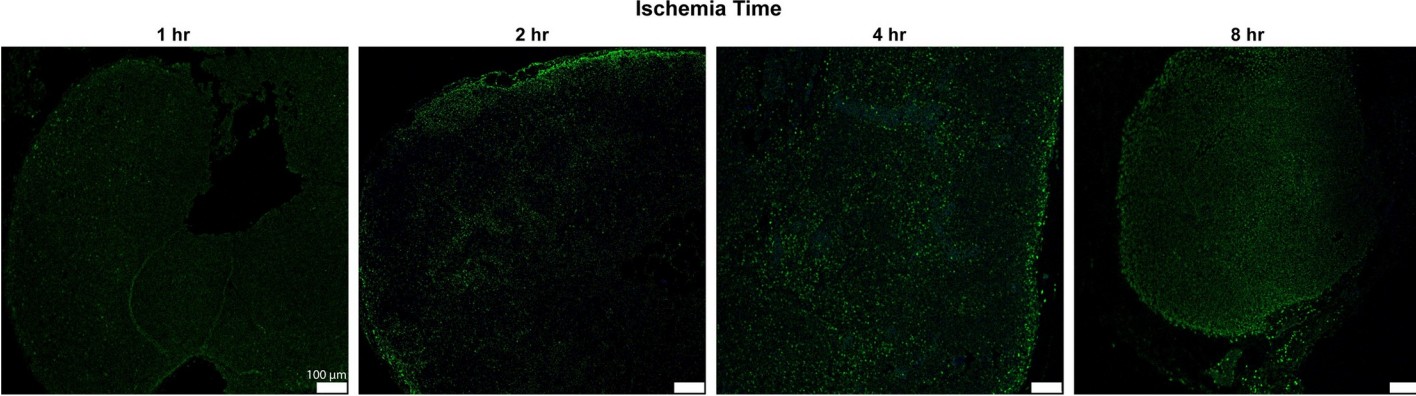

**Fig 9. Histologic and immunofluorescence analysis for TUNEL in rat lymph node cross-sections.** Representative images of lymph node cross sections of TUNEL staining (green) with DAPI (blue) after various ischemia times. x10 magnification, x1.0 zoom. Scale bar = 100 um.

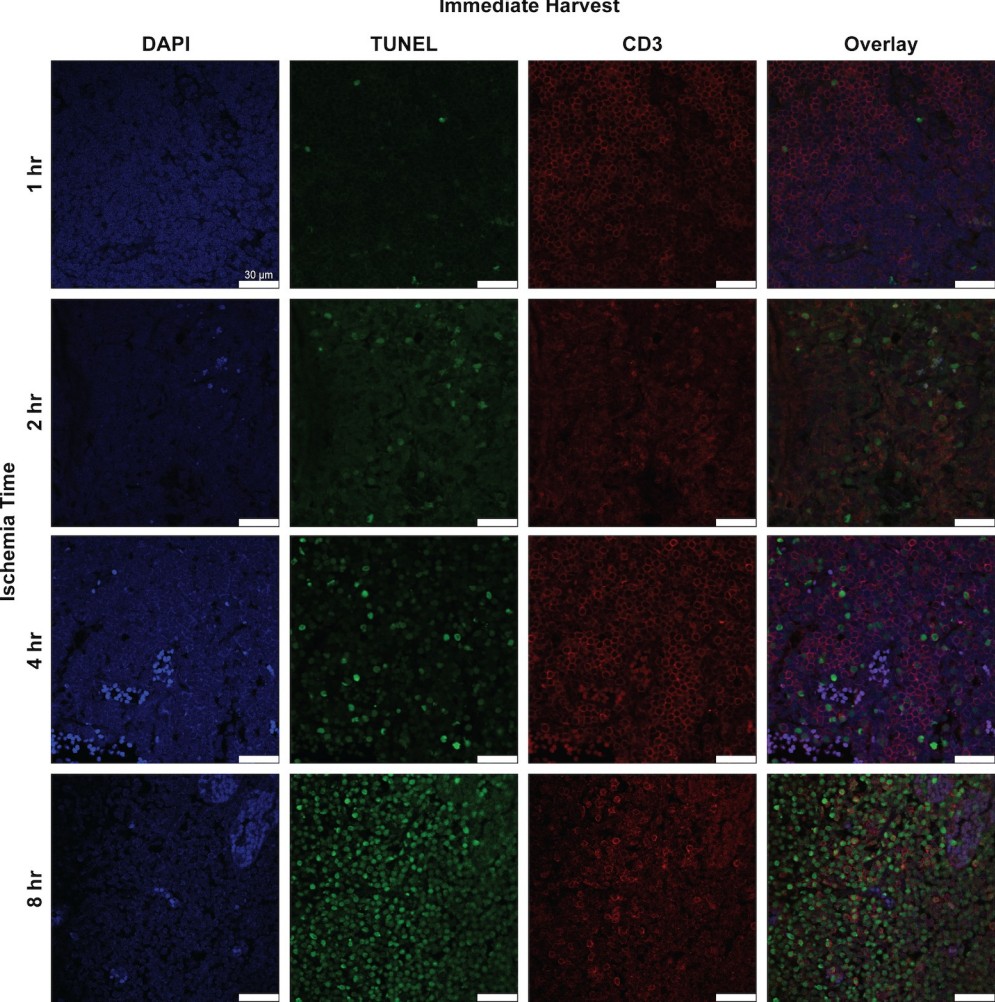

**Fig 10. Histologic and immunofluorescence analysis for CD3+ cells and apoptotic CD3+ cells in rat lymph node cross-sections.** Representative images of lymph node cross sections of double immunofluorescence staining for CD3 (red) and TUNEL (green) with DAPI (blue) after various ischemia times with immediate harvest. x63 magnification, x1.0 zoom. Scale bar = 25 um.

describe changes in ischemia-reperfusion specific gene expression in lymph nodes after ischemia and reperfusion. We also aimed to define a critical ischemia time.

Not surprisingly, and consistent with previous findings, the degree of cell injury and subsequent apoptosis increased with ischemia time [21]. Our data demonstrates a significant increase in the rate of cell apoptosis beginning at 4 hours of ischemia. Additionally, immune cell death increased proportionally with prolonged reperfusion, with significant ischemia-reperfusion injury after 24 hours of reperfusion. This suggests that the immune cell components of lymph nodes are especially sensitive to reperfusion injury. Finally, lymph nodes underwent a significant fibrotic change after 4 hours of ischemia. Intuitively, we would hypothesize that the significant fibrosis of lymph nodes at 4 hours of ischemia would have an impact on the lymphatic drainage, however, functional studies of lymphatic flow are needed.

Upregulation of the proinflammatory cytokines CXCL1/GRO-α, TNF-α, MUC1, and PECAM-1 was as a marker of ischemia reperfusion injury in lymph nodes. Dynamic changes in these cytokines have been previously implicated in ischemia reperfusion injury. For

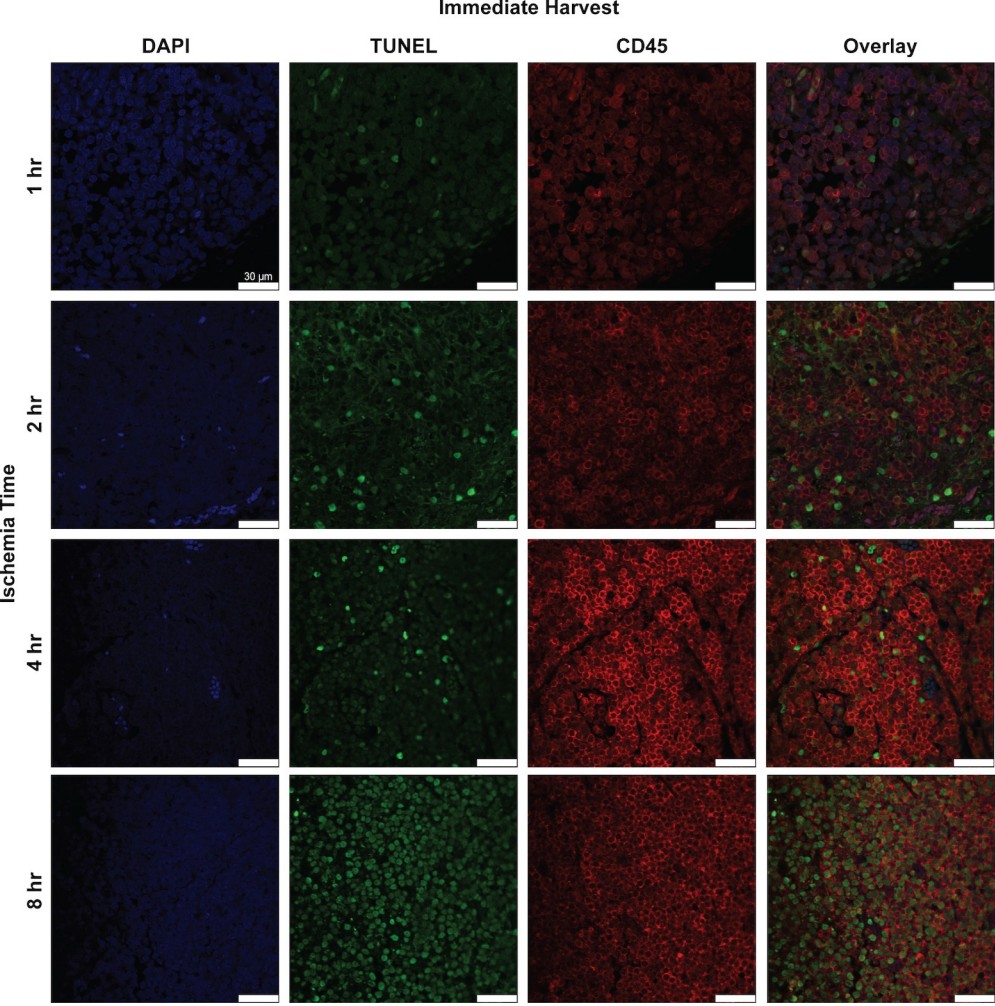

**Fig 11. Histologic and immunofluorescence analysis for CD45⁺ cells and apoptotic CD45⁺ cells in rat lymph node cross-sections.** Representative images of lymph node cross sections of double immunofluorescence staining for CD45 (red) and TUNEL (green) with DAPI (blue) after various ischemia times with immediate harvest. x63 magnification, x1.0 zoom. Scale bar = 20 um.

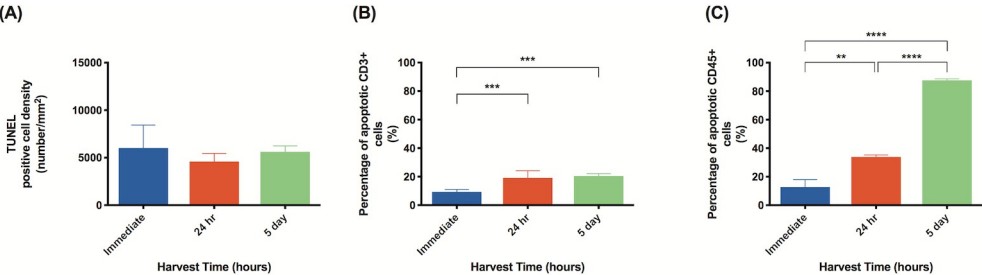

**Fig 12. With an ischemic period of 4 hours, average percentage of apoptotic leukocytes increases with longer harvest time.** Quantitative analysis of: (A) apoptotic cell density in tissue subject to 4 hours of ischemia; (B) percentage of apoptotic T cells in tissue subject to 4 hours of ischemia; (C) percentage of apoptotic leukocytes in tissue subject to 4 hours of ischemia. Analysis was completed in three random high-powered fields per lymph node. Data are presented as mean ± S.E.M. N = 4 lymph nodes per group. *$p < 0.05$, **$p < 0.01$, ***$p < 0.001$, ****$p < 0.0001$.

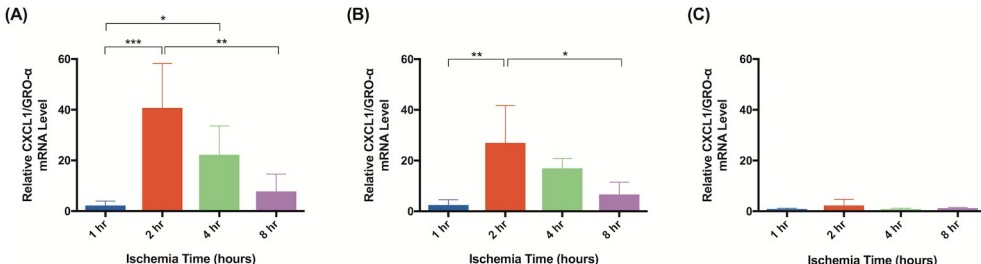

**Fig 13. CXCL1/GRO-α mRNA levels significantly increase after 2 hours of ischemia.** The relative mRNA levels of CXCL1/GRO-α are compared at various ischemia times. Tissue was harvested (A) immediately, (B) after 24 hours of reperfusion, or (C) 5 days of reperfusion. Elevated mRNA levels indicate increased gene expression relative to animals not subject to ischemia. Data are presented as mean ± S.E.M. N = 4 lymph nodes per group. *p<0.05, **p<0.01, ***p<0.001.

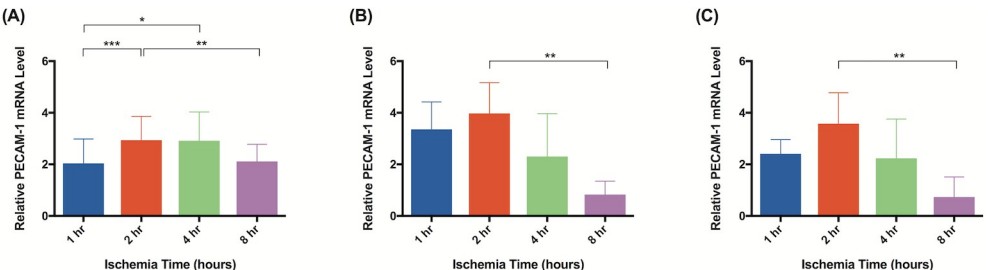

**Fig 14. PECAM-1 mRNA levels increase after 2 hours of ischemia.** Quantitative rtPCR analysis of relative PECAM-1 mRNA levels. Tissue was harvested (A) immediately, (B) after 24 hours of reperfusion, or (C) 5 days of reperfusion. Elevated PECAM-1 mRNA levels indicate increased gene expression relative to animals not subject to ischemia. Data are presented as mean ± S.E.M. N = 4 lymph nodes per group. *p<0.05, **p<0.01, ***p<0.001.

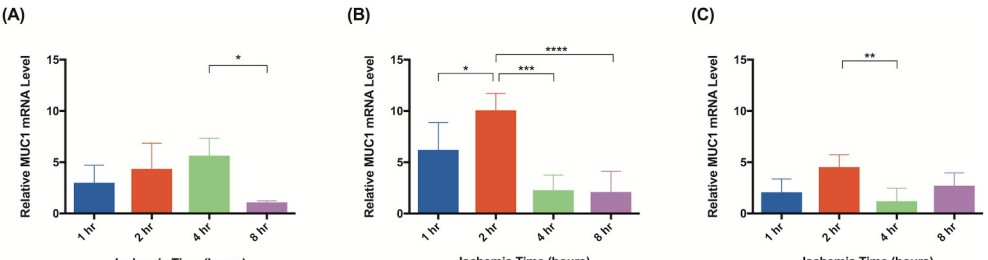

**Fig 15. MUC-1 levels increase shortly after induction of ischemia.** Quantitative rtPCR analysis of relative MUC1 mRNA levels. Tissue was harvested (A) immediately, (B) after 24 hours of reperfusion, or (C) 5 days of reperfusion. Elevated MUC-1 mRNA levels indicate increased gene expression relative to animals not subject to ischemia. Data are presented as mean ± S.E.M. N = 4 lymph nodes per group. *p<0.05, **p<0.01, ***p<0.001.

example, CXCL1/GRO-α is upregulated in the early phases of cerebral ischemia, and promotes leukocyte migration to the inflamed tissue [29]. Enhanced levels of TNF-α expression have been observed following ischemia in both the brain [26, 30] and cardiac tissue [28]. Serum PECAM-1 levels have been shown to be elevated following ischemia reperfusion in rats [33]. Additionally, blocking PECAM-1 attenuated neutrophil migration and accumulation in rat muscle flaps, demonstrating an overall protective effect against ischemia-reperfusion injury [31, 32].

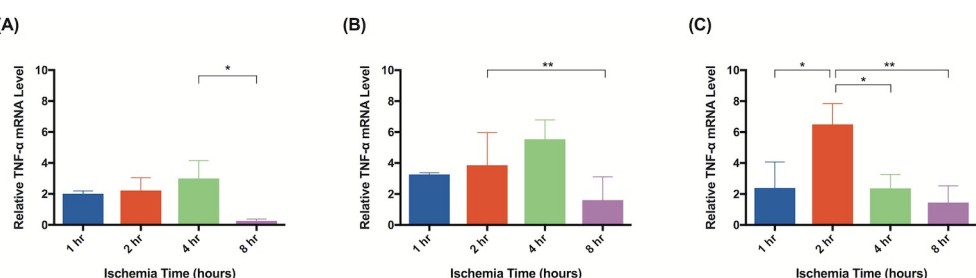

**Fig 16. TNF-α levels are upregulated in response to ischemia.** Quantitative rtPCR analysis of relative TNF-α mRNA levels. Tissue was harvested (A) immediately, (B) after 24 hours of reperfusion, or (C) 5 days of reperfusion. Elevated TNF-α mRNA levels indicate increased gene expression relative to animals not subject to ischemia. Data are presented as mean ± S.E.M. N = 4 lymph nodes per group. *p<0.05, **p<0.01, ***p<0.001.

Subjecting groin flaps to as little as 2 hours of ischemia resulted in significant changes in lymphatic gene expression. A 40-fold transient increase in the expression of CXCL1/GRO-α resulted after just two hours of ischemia. However, prolonged reperfusion (5 days) returned expression levels of CXCL1/GRO-α back to baseline whereas MUC1, PECAM-1, and TNF-α expression remained elevated after the same reperfusion period. CXCL1/GRO-α levels peaked with 2 hours of ischemia in tissue harvested immediately after reperfusion was established. PECAM-1 and MUC1 both peaked with 2 hours of ischemia and 24 hours of reperfusion. TNF-α levels peaked with 2 hours of ischemia and 5 days of reperfusion. These results suggest that CXCL1/GRO-α may be a valuable biomarker to monitor ischemia-reperfusion injury in lymph node flaps due to the dramatic increase in gene expression in response to ischemia.

Understanding the effects of ischemia-reperfusion injury on lymphatic tissue can help guide the perioperative management of patients undergoing VLNT. VLNT is promising therapeutic. A greater understanding of the ischemic tolerance of lymphatic tissue has the potential to improve outcomes and prevent complications [35]. If lymph nodes reach their critical ischemia time intraoperatively, the lymphatic tissue will lose viability. Since the lymph nodes are typically enveloped within fat, which is highly ischemia tolerant, reaching the critical ischemia time of a lymph node may not result in flap loss, but could result in loss of function of the graft. Therefore, a defined critical ischemia time would be a highly practical piece of information. Furthermore, beyond the bedside application of a critical ischemia time, investigating rates of cellular apoptosis, effects on the adaptive immune response, and gene expression of various cytokines may shed light on the mechanisms of reperfusion injury, eventually leading to targeted therapies to block or attenuate the pathways that lead to tissue damage.

## Conclusion

Our data suggest that significant ischemia-reperfusion injury occurs in lymph nodes after 4 hours of ischemia. Thus, vascularized lymph node flaps that are exposed to 4 hours of ischemia during VLNT are at risk for irreversible cellular injury that could impair lymph node flap function. Further exploration of the inflammatory cascade specific to lymph nodes may elucidate useful biomarkers in predicting ischemia-reperfusion injury and, ultimately, flap failure.

## Author Contributions

**Conceptualization:** Ketan M. Patel, Young-Kwon Hong, Alex K. Wong.

**Data curation:** Antoun Bouz, Roy Yu, Sun Young Park.

**Formal analysis:** Cynthia Sung, Roy Yu.

**Investigation:** David P. Perrault, Gene K. Lee, Antoun Bouz, Cynthia Sung, Roy Yu, Austin J. Pourmoussa, Sun Young Park, Wan Jiao.

**Methodology:** Young-Kwon Hong.

**Project administration:** David P. Perrault, Gene K. Lee, Alex K. Wong.

**Software:** Roy Yu.

**Supervision:** Wan Jiao.

**Visualization:** Antoun Bouz, Roy Yu.

**Writing – original draft:** David P. Perrault, Gene K. Lee, Antoun Bouz, Cynthia Sung, Roy Yu.

**Writing – review & editing:** David P. Perrault, Antoun Bouz, Roy Yu, Gene H. Kim, Alex K. Wong.

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
