## [Decision Letter · Decision Letter 0]

31 Oct 2019

PONE-D-19-25885

Ischemia and reperfusion injury in vascularized lymph node flaps

PLOS ONE

Dear Dr. Wong,

Thank you for submitting your manuscript to PLOS ONE. After careful consideration, we feel that it has merit but does not fully meet PLOS ONE’s publication criteria as it currently stands. Therefore, we invite you to submit a revised version of the manuscript that addresses the points raised during the review process.

Please address all comments from reviewers 1 and 3 and my specific questions.. Submit a revised version at your earliest convenience.

We would appreciate receiving your revised manuscript by Dec 15 2019 11:59PM. To enhance the reproducibility of your results, we recommend that if applicable you deposit your laboratory protocols in protocols.io, where a protocol can be assigned its own identifier (DOI) such that it can be cited independently in the future. For instructions see: http://journals.plos.org/plosone/s/submission-guidelines#loc-laboratory-protocols

We look forward to receiving your revised manuscript.

Kind regards,

Cesario Bianchi

Academic Editor

PLOS ONE

Journal Requirements:

2. At this time, we request that you  please report additional details in your Methods section regarding animal care, as per our editorial guidelines:

(1) Please state the source of the rats used in the study  

(2) Please include the method of euthanasia  

Thank you for your attention to these requests.

Additional Editor Comments:

Dear Dr Wong,

Thank you for your submission. Please answer to every reviewers comments and make changes to the manuscript your find necessary.

I have few comments?

- the histology figures are not o f good quality and need to be improved. The magnification is too low power not allowing looking for histological details.

- Since you were interested in studying the lymphatic tissue why you did not use a lymphatic marker as, for example Lyve-1.

Reviewers' comments:

Reviewer's Responses to Questions

**Comments to the Author**

1. Is the manuscript technically sound, and do the data support the conclusions?

Reviewer #1: Yes

Reviewer #2: Yes

Reviewer #3: Yes

2. Has the statistical analysis been performed appropriately and rigorously? 

Reviewer #1: Yes

Reviewer #2: Yes

Reviewer #3: Yes

3. Have the authors made all data underlying the findings in their manuscript fully available?

Reviewer #1: Yes

Reviewer #2: Yes

Reviewer #3: Yes

4. Is the manuscript presented in an intelligible fashion and written in standard English?

Reviewer #1: Yes

Reviewer #2: Yes

Reviewer #3: Yes

5. Review Comments to the Author

Reviewer #1: I commend the authors on a well written paper.

The main concern regards the translational nature of experimental data from the animal model to humans. Another question as stated in the discussion section is how much of the lymph transport is affected by schema time since this was not the scope of the paper.

Reviewer #2: In this experimental study, the authors investigate the effect of ischemia and ischemia-reperfusion on lymph nodes, specifically epigastric lymph node flaps, since very little is known about the effect of ischemia on lymphatic tissue. They evaluated rats’ tissue specimens changes through both biochemical and histological methods and concluded that vascularized lymph node flaps that are exposed to 4 hours of ischemia during VLNT will likely undergo irreversible cellular injury that could impair graft function.

Overall, this is an interesting study as it adds a real contribution to the current literature on VLNF that mainly consists of human case series. The presentation is complete for a scientific paper. It is well-written and free of typographical and grammatical errors. The title reflects the content of the paper; the abstract describes the essential information of the work and the introductory section adequately explains the framework and the problems of the research. Figures and tables are clearly presented.

I applaud the authors for the deep biochemical and histological investigation and the good quality of the pictures. Below are my comments:

Materials and methods

- How was the number of rats to be used in the study calculated?

- Who evaluated tissue changes? Was it a blinded or open label evaluation?

- How was the statistical analysis conducted? Which Software was used?

Please provide full information of the tests that have been used to report the results of the study

- Did you test the normal distribution of your population or did you assume it was not normally distributed?

Reviewer #3: - In this manuscript, authors investigate the cellular changes that occur in lymph nodes in response to ischemia and reperfusion. Lymph node containing superficial epigastric artery-based groin flaps were isolated in a Sprague Dawley rat model. Flaps were subjected to ischemia for either 1, 2, 4, or 8 hours, by temporarily occluding the vascular pedicle. Flaps were harvested after 0 hours, 24 hours, or 5 days of reperfusion. They concluded that significant cell death and changes in tissue morphology don’t occur until after 4 hours of ischemia; however, analysis of inflammatory biomarkers suggests that ischemia reperfusion injury can occur with as little as 2 hours of ischemia.

- This article is well written and well thought-out. I applaud the authors for their effort. However, manuscript needs minor revision. Here are my comments and suggestions:

-Regarding Title, authors should add that this study was made in flaps based on the superficial inferior epigastric artery and vein.

- Figure legends should be removed from the manuscript.

6. PLOS authors have the option to publish the peer review history of their article (what does this mean?). If published, this will include your full peer review and any attached files.

Reviewer #1: No

Reviewer #2: No

Reviewer #3: No

---

## [Author Response · Author response to Decision Letter 0]

14 Dec 2019

Additional Editor Comments:

Dear Dr Wong,

Thank you for your submission. Please answer to every reviewers comments and make changes to the manuscript your find necessary.

I have few comments?

- the histology figures are not of good quality and need to be improved. The magnification is too low power not allowing looking for histological details.

Upon further investigation, it appears that the figures were compressed during the upload process, resulting in poor quality. We reexported the histology figures using the PACE tool recommended by the PLOS One author guidelines, so that they are of higher resolution. The new figures are attached to this email. The current magnification allowed us to visualize individual cells and thus was used for analysis. 

- Since you were interested in studying the lymphatic tissue why you did not use a lymphatic marker as, for example Lyve-1.

We appreciate this very relevant question. To circumvent the need to do confirmatory staining with a marker such as LYVE-1, we utilized Prox-1 eGFP transgenic animals which faithfully express eGFP in lymphatic tissues as previously described by our laboratory in the following publication: 

Jung E, Gardner D, Choi D, Park E, Jin Seong Y, Yang S, Castorena-Gonzalez J, Louveau A, Zhou Z, Lee GK, Perrault DP, Lee S, Johnson M, Daghlian G, Lee M, Jin Hong Y, Kato Y, Kipnis J, Davis MJ, Wong AK, Hong YK. Development and Characterization of A Novel Prox1-EGFP Lymphatic and Schlemm's Canal Reporter Rat. Sci Rep. 2017 Jul 17;7(1):5577. 

In the referenced paper, we previously demonstrated concordance between eGFP signal and lymphatic markers such as LYVE-1 and Podoplanin. Since this model has been well established in our lab and published in the peer reviewed literature, further confirmatory immunostaining with markers such as LYVE-1 was not performed. 

A benefit of the Prox-1 eGFP rat was that realtime GFP fluorescence was used to precisely identify lymphatic tissue at the time of harvest. To increase the precision and clarity of our methodology, we have added specific statements in the manuscript regarding utilization of the Prox-1 eGFP rat.

Review Comments to the Author

2. At this time, we request that you please report additional details in your Methods section regarding animal care, as per our editorial guidelines:

(1) Please state the source of the rats used in the study

 The Prox1-EGFP reporter rats used in this study were developed previously, characterized our laboratory, and described by Jung, E. et. al. [23], Specifically, a Prox1-harboring BAC (RP23-360I16), where the EGFP gene was inserted distal to the mouse Prox1 proximal promoter was used to generate a Sprague-Dawley Prox1-EGFP founder line at Cyagen Biosciences and subsequently expanded at our facility.

(2) Please include the method of euthanasia

 Animals were euthanized via carbon dioxide asphyxiation followed by a confirmatory double thoracotomy.

Reviewer #1: I commend the authors on a well written paper.

The main concern regards the translational nature of experimental data from the animal model to humans. Another question as stated in the discussion section is how much of the lymph transport is affected by schema time since this was not the scope of the paper.

Based on prior published literature, the epigastric flap model upon which the vascularized lymph node flap is based is highly similar to humans from an anatomic standpoint and therefore a suitable platform for translational research. One study investigated the vasculature to the integument covering the ventrolateral abdominal wall in a series of 205 rats. The study revealed that there is a high degree of homology in the anatomy of rats and humans within this region of interest (Casal 2017). The rat model effectively allows for a pedicled flap based on the superior inferior epigastric artery that is ischemic on the distal flap end (Petry 1984). As a result, the model has been used extensively to study ischemia-reperfusion injury, flap necrosis, flap microcirculation, arteriovenous fistula, arterial inflow and venous drainage, and flap prefabrication (Hsu 2018). 

With respect to how one might translate information from the ischemia times tested in our experimental model to clinical practice, these experimental data may be used as a reference for clinicians to make independent case specific decisions regarding lymph node flaps that may have sustained long ischemia times. Our data may be used to the guide the decision making process in the clinical setting but we do not suggest that our data be used as absolute clinical criteria for intraoperative management of lymph node flap transfers. 

The question regarding how ischemia time affects lymph transport is a good one. However, the process of lymph node flap elevation on a single arterial/venous pedicle requires separating the lymph node unit from afferent and efferent lymphatic vessels thereby making it impossible to perform functional studies that explore the effect of ischemia on lymph transport. It might be possible to assess intranodal function in future studies but this would be outside the scope of the present manuscript.

Reviewer #2: In this experimental study, the authors investigate the effect of ischemia and ischemia-reperfusion on lymph nodes, specifically epigastric lymph node flaps, since very little is known about the effect of ischemia on lymphatic tissue. They evaluated rats’ tissue specimens changes through both biochemical and histological methods and concluded that vascularized lymph node flaps that are exposed to 4 hours of ischemia during VLNT will likely undergo irreversible cellular injury that could impair graft function.

Overall, this is an interesting study as it adds a real contribution to the current literature on VLNF that mainly consists of human case series. The presentation is complete for a scientific paper. It is well-written and free of typographical and grammatical errors. The title reflects the content of the paper; the abstract describes the essential information of the work and the introductory section adequately explains the framework and the problems of the research. Figures and tables are clearly presented.

I applaud the authors for the deep biochemical and histological investigation and the good quality of the pictures. Below are my comments:

Materials and methods

- How was the number of rats to be used in the study calculated?

A pilot study was performed in order to determine the mean number of lymph nodes per groin flap in each rat. Each flap generally contained one large lymph node that was suitable for harvest, and rarely two were found per flap. As flaps were created bilaterally in each animal, two nodes were obtained per rat. Based on a G-power calculation and accounting for 20% mortality rate based on prior experience, n=12 animals were required per group in order to achieve statistical significance. 

- Who evaluated tissue changes? Was it a blinded or open label evaluation?

All image analyses were performed by a blinded reviewer. For each analysis, three high-power fields per section were randomly selected and then analyzed by a blinded reviewer. 

- How was the statistical analysis conducted? Which Software was used?

- Please provide full information of the tests that have been used to report the results of the study

- Did you test the normal distribution of your population or did you assume it was not normally distributed?

In all graphs, mean 土 standard deviation values are plotted unless otherwise noted. One-way ANOVA with Tukey’s multiple comparisons test was used to determine differences between groups in GraphPad PRISM8 (GraphPad Software, Inc., La Jolla, CA). p-Values <0.05 were considered statistically significant. In order to test for normality, the Anderson-Darling test was performed in GraphPad PRISM8. All graphs and statistical analyses were generated in GraphPad PRISM8.

Reviewer #3: - In this manuscript, authors investigate the cellular changes that occur in lymph nodes in response to ischemia and reperfusion. Lymph node containing superficial epigastric artery-based groin flaps were isolated in a Sprague Dawley rat model. Flaps were subjected to ischemia for either 1, 2, 4, or 8 hours, by temporarily occluding the vascular pedicle. Flaps were harvested after 0 hours, 24 hours, or 5 days of reperfusion. They concluded that significant cell death and changes in tissue morphology don’t occur until after 4 hours of ischemia; however, analysis of inflammatory biomarkers suggests that ischemia reperfusion injury can occur with as little as 2 hours of ischemia.

This article is well written and well thought-out. I applaud the authors for their effort.

However, manuscript needs minor revision. Here are my comments and suggestions:

Regarding Title, authors should add that this study was made in flaps based on the superficial inferior epigastric artery and vein.

The title has been changed to “Ischemia and reperfusion injury in superficial inferior epigastric artery-based vascularized lymph node flaps”.

Figure legends should be removed from the manuscript.

 Figure legends have been removed from the manuscript.

---

## [Decision Letter · Decision Letter 1]

26 Dec 2019

Ischemia and reperfusion injury in superficial inferior epigastric artery-based vascularized lymph node flaps

PONE-D-19-25885R1

Dear Dr. Wong,

We are pleased to inform you that your manuscript has been judged scientifically suitable for publication and will be formally accepted for publication once it complies with all outstanding technical requirements.

With kind regards,

Cesario Bianchi

Academic Editor

PLOS ONE

Additional Editor Comments (optional):

Dear Dr Wong,

thank you for revising the manuscript that is, at the actual version, acceptable for publication.

Reviewers' comments:

Reviewer's Responses to Questions

**Comments to the Author**

1. If the authors have adequately addressed your comments raised in a previous round of review and you feel that this manuscript is now acceptable for publication, you may indicate that here to bypass the “Comments to the Author” section, enter your conflict of interest statement in the “Confidential to Editor” section, and submit your "Accept" recommendation.

Reviewer #2: All comments have been addressed

2. Is the manuscript technically sound, and do the data support the conclusions?

Reviewer #2: Yes

3. Has the statistical analysis been performed appropriately and rigorously? 

Reviewer #2: Yes

4. Have the authors made all data underlying the findings in their manuscript fully available?

Reviewer #2: Yes

5. Is the manuscript presented in an intelligible fashion and written in standard English?

Reviewer #2: Yes

6. Review Comments to the Author

Reviewer #2: (No Response)

7. PLOS authors have the option to publish the peer review history of their article (what does this mean?). If published, this will include your full peer review and any attached files.

Reviewer #2: No

---

## [Editor Report · Acceptance letter]

30 Dec 2019

PONE-D-19-25885R1 

Ischemia and reperfusion injury in superficial inferior epigastric artery-based vascularized lymph node flaps 

Dear Dr. Wong:

I am pleased to inform you that your manuscript has been deemed suitable for publication in PLOS ONE. Congratulations! Your manuscript is now with our production department. 

With kind regards,

on behalf of

Dr. Cesario Bianchi 

Academic Editor

PLOS ONE